# Morphometrical Identification and Phylogenetic Analysis of Rhinonyssidae (Acari: Mesostigmata) Parasitizing Avian Hosts: New Molecular Data

**DOI:** 10.3390/microorganisms11071783

**Published:** 2023-07-10

**Authors:** Susana A. Sánchez-Carrión, Ivan Dimov, Francisco J. Márquez Jiménez, Manuel de Rojas Álvarez

**Affiliations:** 1Departament of Microbiology and Parasitology, Faculty of Pharmacy, University of Sevilla, Profesor García González 2, 41012 Sevilla, Spain; susarsancar@gmail.com; 2Department of Clinical Anatomy and Operative Surgery Named after Professor M.G. Prives, Pavlov First Saint Petersburg State Medical University, L’va Tolstogo Str. 6-8, Saint Petersburg 197022, Russia; doktordimov@mail.ru; 3Department of Animal Biology, Vegetal Biology and Ecology, Faculty of Experimental Sciences, Universidad de Jaén, 23071 Jaén, Spain; jmarquez@ujaen.es

**Keywords:** mites, molecular systematics, Rhinonyssidae, morphometrical, phylogenetic

## Abstract

Members of the family Rhinonyssidae are tiny hematophagous endoparasitic mites that inhabit the nasal cavities of birds and can cause trauma to their hosts. Traditionally, identifying species in this group has relied on observing their morphometrical characteristics. Nevertheless, determining species within this particular group has become more challenging due to the rising number of newly discovered species. Moreover, the morphometrical traits vary depending on the specific genus or group of species being studied. In this study, the complete internal transcribed spacer ITS1, 5.8S rDNA, and ITS2 regions of the ribosomal DNA from eighteen species of rhinonyssid mites belonging to four genera were sequenced to assess the utility of this genomic region in resolving taxonomic questions in this group and to estimate the phylogenetic relationships among the species. Mites were collected by dissecting the nasal cavities of birds under a stereomicroscope. Specimens used for morphometrical analyses were cleared in 85% lactic acid for 1–48 h and mounted in Hoyer’s medium. Other specimens were preserved at −20 °C for molecular studies. From the data obtained in this study, it can be concluded that a thorough review and an accurate morphometrical identification and determination of the discriminatory traits are needed in this group of mites. Moreover, although the ITS1–5.8S–ITS2 fragment solves different taxonomic and phylogenetic problems at the species level, it would be necessary to test new molecular markers, or even a combination of nuclear and mitochondrial markers or different domains of the nuclear 28S rDNA, to discover a reliable taxonomic situation for rhinonyssids.

## 1. Introduction

Rhinonyssid mites are permanent hematophagous endoparasites that move slowly and inhabit the nasal cavities of birds, taking advantage of the most vascularized areas to feed [1]. Usually, these mites do not cause significant pathological problems in their hosts, although their way of feeding could cause direct damage to birds (Rhinonyssidosis avium disease) [2] and can cause trauma and rupture of the nasal tissues. The family Rhinonyssidae includes around 600 species described worldwide, which are currently grouped into 11 genera [3]: *Larinyssus* [4], *Locustellonyssus* [5], *Mesonyssus* [6], *Ptilonyssoides* [7], *Ptilonyssus* [8], *Rallinyssus* [9], *Rhinoecius* [10], *Rhinonyssus* [11], *Sternostoma* [8], *Tinaminyssus* [12], and *Vitznyssus* [13].

The first studies on these parasites date back to the late 19th century, during which several European researchers mentioned the presence of nasal parasites in birds [14]. Berlesse and Trouessart defined the genus *Ptilonyssus* and the subfamily Rhinonyssinae in 1889 [8]. It was not until 1935 that new species were included, and, years later, the works of Castro and Pereira pioneered a larger and more complete compendium [13]. Strandtmann, in the middle of the 20th century, increased the morphological information on this family of mites [15]. Fain began his studies on rhinonyssids from Europe and Africa in 1957, followed by a huge scientific work on rhinonyssids [16]. Several authors continued to make important contributions to our knowledge of the family Rhinonyssidae in the second half of the last century.

Furthermore, the classification of the family Rhinonyssidae has always been based on the morphometry of the mites found. However, many of the morphological characteristics that have been used for discriminatory purposes are based on characteristics that gradually change among closely related species, with great variability observed in groups of species that are taxonomically and ecologically close. This eventually leads to ambiguous identifications, as well as groupings of species that may be closely related, such as the “sairae” group of *Ptilonyssus* [17], the “melloi” group among the *Tinaminyssus* [18], or the “coniventris” group in the genus *Rhinonyssus* [19]

The new techniques of molecular biology, through the study of different molecular markers, have allowed us to establish relationships between the genetics of populations and distinct aspects of morphology [20], providing a complementary approach to the accurate identification of species in different groups of parasites and establishing phylogenetic relationships among them [21,22].

Despite these facts, few molecular studies on rhinonyssid mites have been carried out. The small size, the great difficulty of identifying the huge number of morphologically similar species, the lack of adequate descriptions and illustration of the species [23], and the disagreement among acarologists regarding the discriminatory morphological features for identifying genera and species may explain the high number of synonymies published by different authors.

Thus, in the case of the family Rhinonyssidae, there have been few molecular studies and few molecular markers have been tested as nuclear markers, such as the ITS1–5.8S–ITS2 fragment and the 28S region, or as mitochondrial rDNA (e.g., cytochrome oxidase I (COI) and 16S). This means that molecular data in the GenBank database regarding different markers and/or the number of rhinonyssid species are scarce [17,24,25].

The aim of the present study was to determine the level of variation among the ITS1–5.8S–ITS2 sequences of several different species and populations, and to analyze the phylogenetic relationships among rhinonyssid mites. For this purpose, our dataset was aligned with other ITS1–5.8S–ITS2 sequences that were obtained previously [25,26]

The resulting phylogenetic tree was analyzed according to the current taxonomy of the family Rhinonyssidae. Our data addressed some taxonomic questions, such as the status of the taxa included in species complexes that comprise very closely related species, and the usefulness of cheliceral morphology for identifying genera.

## 2. Materials and Methods

Eighteen species of rhinonyssid mites belonging to four genera were analyzed using new samples collected for this study in Spain and Russia. Three different populations belonging to the same species and two genera were included, alongside some sequences retrieved from GenBank (Appendix A).

Mites were collected by dissecting the nasal cavities of birds under a stereomicroscope. The specimens used for morphometrical analyses were cleared in 85% lactic acid for 1–48 h (depending on the original opacity) and mounted in Hoyer’s medium. Other specimens were preserved at −20 °C for the molecular studies (Appendix A).

Hosts were obtained in different ways. Hunting birds like *Streptopelia turtur* and *Columba palumbus* were provided to us by hunters. The “Centro Zoosanitario de Sevilla”, which is in charge of the bird population control program developed in the city of Seville, captured the individual birds and provided them to us frozen for parasite examination (e.g., *Columba livia*, *Streptopelia decaocto*, *Passer domesticus*). Most of the remaining surveyed bird species were obtained dead under various circumstances: on the roads, because of high-voltage transmission lines, or even due to adverse weather (elevated temperature and lack of water during the summer, etc.) (Appendix A).

### 2.1. Morphometrical Study and Identification

For an accurate identification of each of the species and populations which was considered, illustrations from the dorsal and the ventral views were made on paper using a Nikon Eclipse Microscope with a camera lucida.

All specimens from each population were analyzed using the discriminatory morphological features considered by different authors. In addition, a brief description of each genus is reported here.

For *Tinaminyssus* spp., the most important morphometrical features for discriminating the species are the shape of the chelicerae, a length of 450–630 µm, the presence and shape of the poststigmatal shields, chaetotaxy and the shape of the dorsal podosomal shield, the arrangement of the z6 setae, the length of the Jv1 and Jv2 setae, the gnathosomal formula, the number of ventral setae and convexitas coxae, and the presence of spina coxae [27,28,29].

*Ptilonyssus* spp. are mites with oblong bodies 380–760 µm long. The dorsal side of the podosoma has only a podosomal shield, the dorsal side of the opisthosoma has only one shield (either opisthosomal or pygidial), and the stigma has oblong peritremes and is located dorsolaterally. There can be presence or absence of the mesosomal shield, an absent poststigmal shield, terminally located gnathosoma, presence or absence of the deutosternal denticles, absence of the tritosternum, presence or absence of the sternal shield, presence of genital and anal shields, an anus with the anal shield located ventrally, the presence of asperos, and absence of the convexitas and spina coxae [3,7,13,16,27,28,30,31].

The *Rhinonyssus* species consists of mites with generally rounded idiosomes, a length of 430–970 µm, the presence of a podosomal shield, the absence of opisthosomal and pygidial shields, dorsolaterally located stigmata without peritrems, presence or absence of mesosomal shields, the absence of a poststigmal shield, terminally located gnathosoma, the absence of deutosternal denticles, the absence of a tritosternum, presence or absence of the primary scutum, the presence of a genital shield, an anal shield that is present or strongly reduced, an anus located on the ventral or dorsal side of the opisthosoma, the presence or absence of asperos, and an absence of convexities and spines on the coxae [7,15,16,27,28,30,32,33].

The *Sternostoma* species is characterized by an oval body and a length of 330–630 µm. On the idiosoma, the two dorsal shields are podosomal and opisthosomal. Pygidial shields are absent. The stigmata are located dorsolaterally, and the peritremes are absent. Mesosomal shields are present or absent. Poststigmal shields are absent. The gnathosoma are located ventrally, and only the most distal part can extend beyond the frontal edge of the idiosoma. Deutosternal denticles are present or absent. The tritosternum is absent. Sternal, genital, and anal shields are present. The anus has an anal shield situated ventrally or terminally. Asperos can be present or absent [3,7,16,27,28,31,32,34].

Moreover, twelve morphometrical measures and two ratios between the measures (LOS/WOS and LPS/WPS) were retrieved. Morphometrical measurements of the dorsal and ventral sides of *Tinaminyssus* spp., *Ptilonyssus* spp., *Rhinonyssus* spp., and *Sternostoma* spp. were made, namely, the length of the body (including palps) (LB), the width of the idiosoma (WI), the length of the podosomal shield (LPS), the width of the podosomal shield (WPS), the length of the opisthosomal shield (LOS), the width of the opisthosomal shield (WOS), the length of the anal shield (LAS), the width of the anal shield (WAS), the length of the ventral view of the gnathosoma (including palps) (LG), and the width of the gnathosoma (WG) [26].

Furthermore, in 2016, Dimov [35] introduced abbreviations and Latin terms for the setae and some other structures, as follows: MS, mesosomal shield (mesosomal scutulum); LMSL, length of the left mesosomal shield; LMSR, length of the right mesosomal shield; WMSL, width of the left mesosomal shield; WMSR, width of the right mesosomal shield; PSS, poststigmal shield; PgS, pygidial shield; Dd, deutosternal denticles; Cc, convexity on the coxa (convexitas coxae); Sc, spines on the coxa (spina coxae); α1–α4, microphytes of the opistosomal shield; dch, chelicera finger; bch, base of the chelicera; An, anus; LPgS, length of the pygidial shield; WPgS, width of the pygidial shield. Some of these features have been considered for morphological identification.

### 2.2. Molecular Study

DNA was isolated from single mites using the NucleoSpin^®^ Tissue XS DNA extraction kit manufactured by Macherey-Nagel (GmbH & Co. KG, Düren, Germany). Briefly, the mites were coarsely crushed with a plastic pestle in a 1.5 mL microcentrifuge tube containing 20 μL of an extraction buffer according to the manufacturer’s protocol. Each sample was checked on SYBR Safe stained with 2% Tris-borate-ethylenediaminetetraacetic acid and 0.8% agarose gel. The PCR mix for amplifying the ITS region consisted of 10 μL of 10× PCR buffer, 1 μL of 10 mM dNTP, 6 μL and 50 mM MgCl_2_, 1.5 μL of 10 μM forward primer, 1.5 μL of 10 μM reverse primer, 10 μL of template DNA, and 0.5 μL of Taq DNA polymerase (2.5 units). The samples were autoclaved with double-distilled water for a final volume of 50 μL.

For amplification of the ITS region, the initial denaturation step was carried out at 94 °C for 2 min, followed by 35 cycles of 20 s at 94 °C for denaturation, 30 s at 53 °C for annealing the primers, and 45 s at 72 °C for primer extension. A final extension step was carried out for 7 min at 72 °C. The primers were as follows: forward, 5-AGAGGAAGTAAAAGTCGTAACAAG-3′; reverse, 5-ATATGCTTAAATTCAGGGGG-3 [36].

The PCR products were sequenced in two directions using the Sanger method by the commercial company Allgenetics (A Coruña, Spain). GenBank accession numbers for the sequenced samples are provided in Appendix A.

#### Sequence Alignments and Phylogenetic Analyses

The sequences of the ITS region were aligned using MEGA v.5.2 [37] and adjusted manually, whereas DAMBE v.5.0 [38] was used to optimize the alignments. Pairwise distance matrices were generated using the Kimura two-parameter model (K2P) [39].

Phylogenetic trees were inferred on the basis of the nucleotide data and built using two methods. Maximum likelihood (ML) trees were generated using PHYML v.3.0 [40], and Bayesian inference (BI) trees were built using Mr. Bayes v.3.2.6 [41]. JMODELTEST 2.1.10 [42] was used to select the best-fitting model of nucleotide substitution for the alignment of the ITS region. Models of sequence evolution were chosen for the subsequent analyses according to the Akaike Information Criterion [43,44]. The best-fitting nucleotide substitution models selected by JMODELTEST included a general time-reversible model with a gamma-distributed rate variation of GTR + G for the ITS region. These nucleotide substitution models were used to analyze the phylogenetic relationships using PHYML. Support for the topology was examined by bootstrapping the original dataset 1000 times (the heuristic option) [45]. The best-fitting nucleotide models selected by JMODELTEST and available for BI in MrBayes were GTR + G (nst = 6 with gamma rates) For BI, the standard deviation of the splitting frequency was used to assess convergence; the chains were sampled every 500 generations and each dataset was run for 10 million generations. Burn-in was determined empirically by examining the log-likelihood values of the chains. Phylogenetic trees based on the ITS region were rooted with a species of the superfamily Dermanyssoidea, namely, *Dermanyssus gallinae*.

## 3. Results

### 3.1. Morphometrical Identification

All mites were identified strictly on the basis of the descriptions, measurements, and the best illustrations and redescriptions of females that different authors have carried out, with which the specimens considered in this study were compared. Thus, a summary of the most important discriminatory traits is reported in Appendix A.

Previous works were used to identify the species [3,6,7,13,15,16,18,19,27,27,31,32,46,47,48,49,50,51,52,53,54,55,56,57,58,59,60,61,62,63]. The specific publications are presented in Appendix A within the descriptions of each identified species.

#### Molecular Results

There was little variability in the lengths of the different sequences, which ranged between 449 and 536 bp, with an average of 506 bp.

The nucleotide composition (Appendix A) showed a higher content of adenine and thymine (29.6% and 31.7%, respectively), while cytosine had the lowest concentration (14.5%), followed by guanine (24.2%). In the collection of these data, the values of the external elements were not considered.

Table 1 shows the fragment lengths (in base pairs) of the ITS1, 5.8S, and ITS2 fragments. The boundaries were determined by using the conserved areas flanking the 18S, 5.8S, and 28S sequences according to Navajas et al. [36] and de Rojas et al. [64].

Thus, when the fragments of the ITS1–5.8S–TS2 region of the species considered in the present study were aligned, an alignment was obtained with 576 positions, including gaps, with 197 conserved and 353 variables.

The pairwise distance matrix produced by a pairwise comparison of all considered taxa is given in Appendix A, which also includes the percentages of similarity between the taxa. Table 2 shows a summary of the intergeneric, interspecific, and intrapopulation distances, as well as the percentage of similarity.

### 3.2. Phylogenetic Study of the Species Included in the Alignment

In the analysis of the sequences aligned with Mega version 11.01 and Clustal W version 2.0, an alignment was obtained for the ITS1–5.8S–ITS2 fragment of 576 base pairs, finding 197 preserved positions and 353 variables.

Phylogenetic trees were constructed as explained in the Materials and Methods section. The topology of the Bayesian tree, including all the considered taxa with *Dermanyssus gallinae* as an outgroup, is shown in Figure 1. Each node in this tree shows the Bayesian posterior probabilities (PP) followed by the ML bootstrap values (BS).

## 4. Discussion

The question of delimiting species in molecular genetics remains a problem to be solved because there is no objective limit which indicates that two species are distinct [24]. However, Dobler and Farrell [65] observed that, in the case of parasitic organisms, “gene flow between parasite populations in different host species may be limited, even if the parasites are not 100% specific”. This structure can favor the formation of races or speciation.

The family Rhinonyssidae is believed to have descended from ectoparasitic ancestors related to the Macronyssidae [4]. Rhinonyssid genera vary in their degrees of host specificity, with some genera being restricted to one host family and others found in hosts from different orders [48]. In North America, genera with broad host ranges are *Ptilonyssus* (passeriform, caprimulgiform, falconiform, and apodiform hosts), *Rhinonyssus* (anseriform, podicipediform, and charadriiform hosts), *Sternostoma* (passeriform, piciform, and charadriiform hosts), and *Tinaminyssus* (ciconiiform and columbiform hosts). Among the more host-specific rhinonyssids, *Rhinoecius* species parasitize owls (Strigiformes), *Rallinyssus* species parasitize rails (Rallidae), and *Larinyssus* species parasitize gulls and terns (Laridae) [3].

### 4.1. Morphological Analysis

Throughout the course of their study, the principles governing the construction of the natural rhinonyssid system have undergone significant changes. These changes have often required a complete review of the supraspecific classification, with the most challenging aspect being the development of criteria for determining the genera and subfamily ranks, such as the morphology of the chelicerae (Figure 2). Thus, firstly, an accurate identification of the studied species is needed. In the Appendix A, a brief description including all discriminatory characteristics of the species identified in this study is shown.

It is noteworthy that, with regard to the genus *Tinaminyssus*, in 1948, Castro [13] described *Neonyssus melloi* as a new species from some specimens found in the nasal cavities of *Columba livia* domestica L. According to this author, the new species was characterized by its very strong legs with highly developed nails, a gnathosome that was small in size in relation to the body, chelicerae of uniform diameter, and peritremal scutes surrounding the stigmata.

In 1960, Fain [6] created the genus *Mesonyssus* to include the *Neonyssus* species endowed with long cheliceral fingers. Therefore, *Neonyssus melloi* became part of the new genus *Mesonyssus*. However, in 1958, Strandtmann and Wharton [9] had defined the genus *Tinaminyssus* on the basis of a series of characteristics, including all those assigned by Fain to the genus *Mesonyssus*. Therefore, the generic name proposed by Strandtmann and Wharton had priority, as indicated by Domrow in 1969 [27], who considered that the correct name for the genus should be *Tinaminyssus*.

### 4.2. Molecular Analysis

As shown in the Appendix A, the length of ITS1 varied between 232 bp in *Sternostoma fulicae* and 193 in *Rhinonyssus tringae*. The 5.8S rDNA gene ranged from 189 bp in *Ptilonyssus hirsti* to 177 in *Rhinonyssus echinipes*. The ITS2 fragment was less variable in size, with all species showing values very close to 90. These values are very similar to those found by de Rojas et al. [64] and Úbeda et al. [66]. In addition, according to the GenBank database, members of the Dermanysoidea superfamily (*Dermanyssus longipes* FM179377 and *Androlaelaps casalis* AM903317) showed similar values for ITS1 and for the 5.8s rDNA gene, with the ITS2 region being significantly shorter (73 and 62 bp, respectively).

The contents of the different bases are shown in Appendix A as percentages. As can be seen in Appendix A, our results were practically identical to those obtained by de Rojas et al. [64] for a smaller number of rhinonyssid species.

Likewise, the lengths of the markers considered in the different species studied herein, as shown in Appendix A, were also similar to those obtained by Úbeda et al. [66] for species of the genus *Rhinonyssus*, ranging from 468 bp for *R. himantopus* to 506 in *R. neglectus* to 540 bp for *Sternostoma boydi* [64], as well as to those of other representatives of the superfamily Dermanysoidea, such as *Dermanyssus longipes* (accession number FM179377).

With regard to the variability of the different fragments (Table 1), as expected, the 5.8S rDNA gene was the most conserved region, while the ITS1 region was the longest and most variable. Finally, ITS2 was the shortest and, unusually, quite preserved fragment, at least among species from the same genus. This fact was observed by Navajas et al. [36] in mites of the family Phytoseiidae, and by de Rojas et al. [64] in representatives of the Rhinonyssidae.

### 4.3. Analysis of Phylogenetic and Genetic Distances

It was not until 1979 that Pence attempted to address phylogeny in this family of mites. Pence, on the basis of Castro’s observations [13], hypothesized that the chelicerae would have fairly stable morphological characteristics, since they are related to nutrition. According to this hypothesis, the rhinonyssid mites would come from some ancestor of the Macronysidae in which the morphology of the chelicerae had some similarity to those of the genus *Tinaminyssus* [67]. Figure 2 shows drawings of the chelicerae of all species included in this study.

Thus, according to the phylogenetic tree created as explained in the Materials and Methods sections and shown in the results (Figure 1), several questions can be answered.

The topology of the tree shown in Figure 1 corresponds to that created via the Bayesian inference method, and is similar to that obtained via the maximum likelihood method. However, in the latter case, the bootstrap values were slightly lower than the Bayesian probabilities. In both cases, the four genera considered in this study were shown to be in different clades. This observation coincides with the morphological proposal of Pence (1979) [68] regarding the classification of genera based on the morphology of the chelicerae. However, species of the genus *Sternostoma* appeared in a separate branch, and the species of the other three genera were gathered together in a polytomy, which means that a more conserved marker must be tested to resolve the phylogeny at the genus level.

The clade that includes the species of the genus *Tinaminyssus* appears in a common branch with Bayesian posterior probabilities (PP) and ML support (BS) of 100/99. Within this group, two different branches can be distinguished, depending on the morphological characteristics: the “melloi” complex (*T. melloi, T. streptopelioides, and T. streptopeliae*) and the “columbae” complex (*T. minisetosum* and *T. columbae*) [18]. However, *Tinaminyssus bubulci*, a parasite of pelecaniform birds, appears in the “melloi” group with the species parasitizing columbiform birds (100/99). Furthermore, two of the morphological criteria for including a species in the “columbae” group are a lack of postanal setae and barely developed opisthosomal setae. *T. bubulci* shows numerous well developed opisthosomal setae (typical of the “melloi” group), but lacks the postanal setae. Thus, this apparent morphological and ecological contradiction must be tested by assaying more isolated or concatenated molecular markers in order to resolve it.

Moreover, *T. streptopeliae* from and *T. streptopelioides* show very similar morphologies and measurements that overlap. They are gathered together (100/92), and the genetic distance between them (0.11) is similar to that between *T. streptopeliae* and *T. melloi* (0.14), which are different species. This agrees with de Rojas et al. [26], who considered the former species to be a cryptic species.

Finally, *T. columbae* and *T. minisetosum*, which were included in the “columbae” group, were gathered together with high support (100/99). However, although these two species were considered to be different species, the genetic distance was the lowest among the species of *Tinaminyssus* (0.05).

The genus *Ptilonyssus* is the most numerous in the family Rhinonyssidae. Currently, there is no consensus among acarologists about its taxonomic boundaries nor the number of species which should be included [31]. According to the morphological signs, some species of the genus have been grouped into species complexes on the basis of the chaetotaxy of the podosomal shield on the dorsal aspect of the idiosoma, namely, the “hirsti”, “lanii”, “motacillae”, “orthonychus”, “pari”, and “sairae” groups [68]. This study included three species belonging to the “hirsti” complex, three belonging to the “sairae” complex, and one belonging to the “pari” complex.

The topology of the Bayesian phylogenetic tree included two sister branches with high to medium support (89/61) that contained species of the “hirsti” complex and the “sairae” complex. The former was supported with 100/99, and gathered species with chelicerae without widened bases. Within this group, two different populations of *P. hirsti* were gathered together with high support (100/100). The other branch gathered *P. fringillae* and *P. pari* from Russia (100/100). In this case, the genetic distance between these species was 0.1, even lower than that between populations of *P. hirsti*. However, *P. pari* from *Parus atricapillus* was described by Fain and Hyland [53] on the basis of pairwise comparisons among *P. capensis, P. emberizae, P. chloris,* and *P. hirsti*. In 1974, Fain and Sixl [52] described a new species: *P. fringillae*. When drawings of the specimens were compared, these species showed the same number of shields, but with slight differences in their development. Thus, two of the dorsal opisthosomal seta were included in the opisthosomal shield in *P. fringillae,* but not in *P. pari*. Furthermore, *P. pari* has two poststigmatic shields that are not shown in *P. pari*. However, the differences in the development of the shields depend on the specimen and the time of collection, as in *T. streptopeliae* and *T. streptopeliodes* [34]. Thus, these traits are not reliable discriminatory characteristics. Therefore, further molecular studies are needed in order to clarify the statuses of *P. fringillae* and *P. pari*.

The second branch grouping *Ptilonyssus* species included *P. motacillae*, *P. motacillae phoenicuri,* and *P. muscicapae*, which show chelicerae that become wider at the base. All these species belonged to the “muscicapae” group, and their clustering was well-supported (100/99). It is noteworthy that the genetic distance between *P. motacillae phoenicuri* and *P. motacillae* was 0.08; currently, these are considered to be the same species. However, the genetic distance between *P. motacillae* and *P. muscicapae* was not much higher (0.10).

All of these data explain the complexity of delimiting species and determining discriminatory traits in groups of closely related species of the family Rhinonyssidae, and further morphological and molecular studies are needed to establish the statuses of species and subspecies within this group.

The last clade in the polytomy, similar to those discussed previously, was well-supported (98/90) and included species of the genus *Rhinonyssus*, most of which parasitize birds from frequently wet and marshy habitats [69]. In this study, the authors established three species groups within this genus: the “coniventris” group, the “himantopus” group, and the “rhinolethrum” group.

Species of the genus *Rhinonyssus* were also grouped into a well-supported common trunk (98/90) with two branches. One branch included species belonging to the “coniventris” complex, while the other included *R. himantopus*, which belonged to the “himantopus” group. Our molecular results were consistent with the morphological data, as *R. himantopus*, which did not belong to the “coniventris” group, had a very different morphology. Moreover, the distances between *R. himantopus* and the other species of the genus *Rhinonyssus* were very high, similar to those obtained between *R. himantopus* and species from any other genera, ranging from 0.45 to 0.48. For this reason, it would be interesting to study more species from the “himantopus” group, because the cheliceral morphology is slightly different from that of the “coniventris” group.

The most similar closely related species in the family Rhinonyssidae were placed in the “coniventris” group, which Fain established in 1963 to include a series of species that were causing confusion and controversy among Rhinonyssidae specialists. The species that gave its name to the complex is *Rhinonyssus coniventris* [11].

Individuals of these species have a single poorly sclerotized dorsal shield that, in certain cases, has become so reduced that it practically only includes the muscular insertion areas characteristic of this region of the idiosoma. The sternal shield does not exist as such, and the genital shield is almost always much longer than it is wide. No anal shield can be observed. In the present study, numerous specimens of mites were located and studied, and their morphological characteristics allowed them to be included in the species *Rhinonyssus echinipes* [57], *R. neglectus* [57], and *R. tringae* [54]. These three species show few morphological differences.

In fact, in 1951, Strandtman [15], discussing the *Rhinonyssus* species described by Hirst, indicated that it was quite probable that a re-examination of the types of *R. neglectus* [57] and *R. echinipes* [57], in the light of abundant material from many hosts, would show that they were conspecific with *R. coniventris* [11]. Later, in 1956, the species *R. neglectus* [57] and *R. echinipes* [57] were considered to be synonyms of *R. coniventris* [34].

In 1963, Fain concluded that: *R. coniventris* is a polytypic species that includes three subspecies (*R. coniventris coniventris, R. c. neglectus* and *R. c. echinipes*), each restricted to one or several closely related genera of Charadriiformes [54].

Finally, in 1984, Butenko [32] concluded that the “coniventris” group included five forms that were assigned species status: *R. coniventris* [11], *R. echinipes* [57], *R. neglectus* [57], *R. shcherbininae* [32], and *R. tringae* [54].

In this study, species of the “coniventris” group were gathered together in a well-supported common branch (100/99). However, it is noteworthy that the genetic distances among these three species were 0.07, 0.08, and 0.10, which are similar to those between *P. motacillae* and *P. motacillae phoenicuri* (0.08), or between *P. motacillae* and *P. muscicapae*, which are very similar (0.10).

Two species of this complex group, *R. neglectus* and *R. tringae*, were grouped together and separated from *R. echinipes*. This branch was well-supported in terms of PP (94) and had medium ML bootstrapping support (64).

Furthermore, the grouping of *R. tringae* and *R. neglectus* in the same branch could be explained by the preference of these two species for Charadriiformes hosts of the Scolopacidae family (*Tringa totanus* and *Calidris alpina*, respectively). In contrast, *R. echinipes* is only found in Charadriiforme hosts of the Charadriidae family.

The combination of the molecular analyses and morphological results suggests that the taxonomic statuses of *R. echinipes* [57], *R. neglectus* [57], and *R. tringae* [54] can be accepted as species, although it would be convenient to prove this by means of other molecular markers. Thus, the molecular data confirm our morphological and ecological observations.

Finally, the only branch grouping species from the same genus, which was not present in the polytomy and was well-supported (100/99), included two species of the genus *Sternostoma.*

In this study, only two species that parasitize aquatic birds could have been included. *Sternostoma boydi* from *Larus argentatus* and *S. fulicae* from *Fulica atra*. These two species are easily distinguishable on the basis of their morphological traits. However, the genetic distance (0.10) is similar to that of closely related species. This fact could indicate a case of phenotypic plasticity for adaptation to very distantly related birds, namely, the charadriformes and gruiformes.

From all of these data, it can be concluded that, although the ITS1–5.8S–ITS2 fragment solves different taxonomic and phylogenetic problems at the species level, it would be necessary to test new molecular markers that resolve the intergeneric relationships and to concatenate two or more molecular markers to resolve the identification of species in complexes that include very closely related species. Even the combination of nuclear and mitochondrial markers or different domains of the nuclear 28S rDNA would be convenient for solving these situations. Furthermore, a thorough review and an accurate morphometrical identification and determination of the discriminatory traits are needed for this group of mites.

## Figures and Tables

**Figure 1 microorganisms-11-01783-f001:**
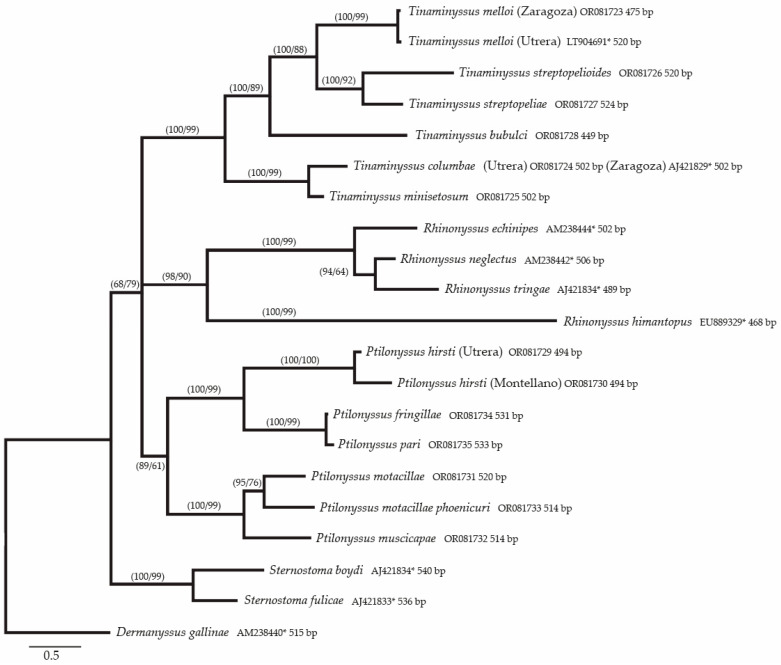
Phylogenetic tree (BI topology) of rhinonyssid mites of different species and geographical origins based on the ITS1-5.8S-ITS2 fragment. Values next to the nodes indicate the Bayesian posterior probabilities (PP), followed by the ML bootstrap values (BS). Support is only indicated on nodes with PP > 0.95 and BS > 75%. Each species shows the GB access number and the length of the amplified fragment. (*) Sequences were retrieved from GenBank.

**Figure 2 microorganisms-11-01783-f002:**
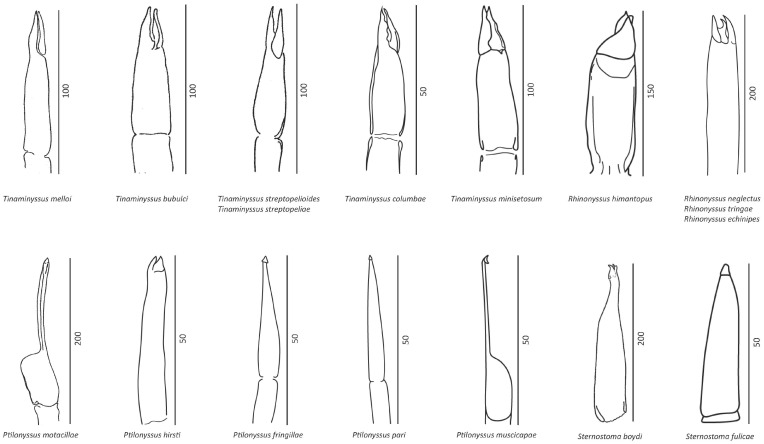
Morphology of the chelicerae of the species identified in this study. Scales are shown in µm.

**Table 1 microorganisms-11-01783-t001:** Lengths of transcribed internal spacers 1 and 2 (ITS1, ITS2) and the rDNA gene 5.8 (in base pairs).

Species	ITS1	5.8S	ITS2
*Tinaminyssus melloi* (Zaragoza) Castro, 1948	218	184	-
*Tinaminyssus melloi* (Utrera) Castro, 1948	218	182	90
*Tinaminyssus columbae* (Utrera) Castro, 1948	200	182	90
*Tinaminyssus columbae* (Zaragoza) Castro, 1948	200	182	90
*Tinaminyssus minisetosum* Butenko, 1984	201	181	90
*Tinaminyssus streptopelioides* Butenko, 1984	218	182	90
*Tinaminyssus streptopeliae* Fain, 1962	218	182	94
*Tinaminyssus bubulci* Zumpt & Till, 1955	198	181	-
*Ptilonyssus hirsti* (Montellano) Castro, 1948	214	189	-
*Ptilonyssus hirsti* (Utrera) Castro, 1948	214	189	87
*Ptilonyssus motacillae* Fain, 1966	214	191	82
*Ptilonyssus muscicapae* Bregetova, 1970	213	189	82
*Ptilonyssus motacillae phoenicuri* Fain, 1966	213	189	82
*Ptilonyssus fringillae* (Montellano) Fain & Sixl, 1971	226	188	87
*Ptilonyssus pari* (Russia) Fain & Hyland, 1963	226	188	89
*Rhinonyssus echinipes* Hirst, 1921	206	177	89
*Rhinonyssus himantopus* Strandtmann, 1951	197	181	90
*Rhinonyssus neglectus* Hirst, 1921	210	177	89
*Rhinonyssus tringae* Fain, 1963	193	177	89
*Sternostoma boydi* Strandtmann, 1951	236	184	90
*Sternostoma fulicae* Guevara & Úbeda, 1975	232	184	90

**Table 2 microorganisms-11-01783-t002:** Variation in the genetic distances using Kimura’s two-parameter model and the percentage of intergeneric, interspecific, and intrapopulation similarity (*, genetic distance between *P. fringillae*–*P. pari*).

Level	Genetic Distance (K2P)	%
Genera		
*Tinaminyssus–Sternostoma*	0.26–0.34	77.6–81.1
* Tinaminyssus–Rhinonyssus *	0.33–0.54	70.5–79.5
* Tinaminyssus–Ptilonyssus *	0.22–0.39	73.8–82.8
* Sternostoma–Rhinonyssus *	0.32–0.42	72.7–77.8
* Sternostoma–Ptilonyssus *	0.23–0.32	76.4–82.5
* Ptilonyssus–Rhinonyssus *	0.31–0.48	71.7–79.3
Species		
* Tinaminyssus *	0.05–0.27	82.5–96
* Sternostoma *	0.1	91.5
* Ptilonyssus *	0.08–0.30 (0.01 *)	72.6–94.1
* Rhinonyssus *	0.08–0.48	78.3–93.2
Populations		
* T. melloi * (Zaragoza–Utrera)	0	100
* T. columbae * (Utrera–Zaragoza)	0	100
* P. hirsti * (Montellano–Utrera)	0.04	96.7

## Data Availability

Sequences of each of the species analyzed in this study are available from NCBI (GenBank accession numbers: from OR081723 to OR081735).

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
