# Peer review of "Morphometrical Identification and Phylogenetic Analysis of Rhinonyssidae (Acari: Mesostigmata) Parasitizing Avian Hosts: New Molecular Data"

_microorganisms, 2023, doi:10.3390/microorganisms11071783_

Round 1

Reviewer 1 Report

The paper is devoted to an such interesting topic of parasitological research as an invasive birds species. Results of integrative methods of investigation of  Rhinonyssidae parasites are new and original. Although the authors have expanded their data since the some previous publication (e. g. DOI: 10.1007/s00436-006-0356-z), they have not fully presented it here. In spite of the high level results obtained by the authors, the manuscript text requires the addition of important information (listed below)  to results be evaluated by the readers.

Therefore, manuscript is suitable to be published in the Journal "Microorganisms" after change as follow:

1.      The Introduction needs to be slightly modified: paragraph on lines 72-77 need to be moved above. Consider to make it as the second paragraph. The references should be re-numbered as well.

2.      The Materials and Methods section should be expanded.

·           Please first write down which sequences were obtained in this study and which were obtained earlier and give references for them (in Table S1).

·           Also indicate not only the place of sampling, but also the country and vice versa. Verify all GB access numbers.

·           Please clarify the number of investigated host of every species.

·           How were the hosts preyed upon?

·           Have any mites species been observed in the same host species?

·           Specify the infestation rates for every parasite species.

·           You pointed that single specimen of every species was used for PCR? Clarify, are all of them hologenophores? If different specimens were used for morphological and genetic analyses, please indicate this.

·           How many specimens were used for the morphological description?

3.       Section Results also need some explanations.

·           Authors didn't include all available sequences (e.g. Tinaminyssus melloi). What the reason? I guess you should add as much s possible sequences of the species mentioned in manuscript even if they are the identical.

·           GB access numbers should be added to the tree on Figure 1. Please indicate the sequences obtained in current study.

·           The "ITS" fragment should be clarify in the capture of Figure 1. Here the length of the fragment should be pointed as well.

·         The Figure 2 should be mentioned in the section 4.1 also?

·         Paragraph on lines 317-329 need to be moved above, to the morphological part (4.1) of Discussion. All these taxonomical reconstructions are based on morphology only.

Also some comments, questions, and corrections you can find as trackers in revised manuscript document.

The manuscript is well written.

Reviewer 2 Report

This work opens a field to developing a DNA barcode for rhinonyssid mites that inhabit the nasal cavities of birds. Due to the importance of birds as bioindicators for ecosystem health, I consider publishing works such as this one very appropriate. The authors made a great contribution, such as the brief description of all discriminatory characteristics of the rhinonyssid species identified by the authors, as well as the contribution of genetic markers to characterize and identified these mites.

Author Response

The reviewer 2 does not submitted reviewer´s comments.

Round 2

Reviewer 1 Report

The authors substantially revised their manuscript and improved its quality. The text and its structure is much clearer. However, there are still a couple of things requiring their attention.

1. Please check the italics throughout the text. In some cases (e.g. Table 2, Fig.1) it is not necessary! Only for species names!

2. Table S1: Please unify the writing of the GenBank Access numbers either all with '.1' or all without '.1'.

In any case, I appreciate the hard work of the author's collective, because Rhinonyssidae is really difficult group of parasites as regards taxonomy, and I congratulate to their manuscript.

Author Response

Please see the atachment
